# Techniques for the Thermodynamic Consistency of Constitutive Equations

Angelo Morro [1],*[ID] and Claudio Giorgi [2][ID]

1 Dipartimento di Informatica, Bioingegneria, Robotica e Ingegneria dei Sistemi, Università di Genova, Via All'Opera Pia 13, 16145 Genova, Italy

2 Dipartimento di Ingegneria Civile Ambiente Territorio Architettura e Matematica, Università di Brescia, Via Valotti 9, 25133 Brescia, Italy

* Correspondence: angelo.morro@unige.it

**Abstract:** The paper investigates the techniques associated with the exploitation of the second law of thermodynamics as a restriction on the physically admissible processes. Though the exploitation consists of the use of the arbitrariness occurring in the Clausius–Duhem inequality, the approach emphasizes two uncommon features within the thermodynamic analysis: the representation formula, of vectors and tensors, and the entropy production. The representation is shown to be fruitful whenever more terms of the Clausius–Duhem inequality are not independent. Among the examples developed to show this feature, the paper yields the constitutive equation for hypo-elastic solids and for Maxwell–Cattaneo-like equations of heat conduction. The entropy production is assumed to be given by a constitutive function per se and not merely the expression inherited by the other constitutive functions. This feature results in more general expressions of the representation formulae and is crucial for the compact description of hysteretic phenomena.

**Keywords:** thermodynamic consistency; constitutive equations; exploitation of the second law; rate equations

**MSC:** 74A15; 74B20; 74J30; 80A17

## 1. Introduction

The thermodynamic consistency of models in continuum physics is established by the compatibility with the balance equations (mass, linear and angular momentum, energy, and entropy). The balance of entropy is expressed by the Clausius–Duhem (CD) inequality ([1], §257). The conceptual role of the CD inequality is now due to a famous paper by Coleman and Noll [2]. Thermodynamic processes are the sets of pertinent fields that describe a body and satisfy the balance equations and the constitutive assumptions. The CD inequality is assumed to hold for all admissible processes and hence it places restrictions on the constitutive assumptions. Hence, the CD inequality, while being equivalent to the assertion that the entropy production cannot be negative, sets restrictions on physical, real processes.

Let $\theta$ be the absolute temperature, $\eta$ the specific entropy, $\mathbf{q}$ the heat flux, and $r$ the energy supply. The physical content of the balance of entropy ([3], §6.5),

$$\text{entropy change} = \text{entropy transfer} + \text{entropy production}$$

is made formal in [2] by letting the entropy transfer consist of the entropy flux $\mathbf{q}/\theta$ and the entropy supply $r/\theta$. Hence, in local form, the entropy production (per unit mass) $\gamma$ is assumed to be given by

$$\gamma = \dot{\eta} - \frac{r}{\theta} + \rho^{-1}\nabla \cdot (\mathbf{q}/\theta), \tag{1}$$

where the superposed dot denotes the total time derivative and $\rho$ is the mass density. The basic postulate in [2] is that the entropy production $\gamma$ is non-negative for every admissible

process. Next Müller [4] observed that the entropy flux, say **j**, need not be **q**/$\theta$. Hence, the admissible constitutive functions of $\eta$ and **q** (or **j**) are required to satisfy $\gamma \geq 0$. Later on, Green and Naghdi [5] pointed out that also the entropy production $\gamma$ has to be considered as given by an admissible constitutive equation. To our knowledge, this view has not received much attention in the literature; among the few approaches involving $\gamma$ as a constitutive function we mention ref. [6], recent works of ours [7–9], and the systematic procedure developed in [10].

The purpose of this paper is two-fold. The first fold is to show the general role of the entropy production. Almost always the constitutive function of $\gamma$ is merely inherited by the other constitutive equations or Equation (1) is an identity. As a simple example, for a rigid heat conductor obeying Fourier's law

$$\mathbf{q} = -\kappa \nabla \theta$$

the CD inequality becomes the heat conduction inequality ([11], § 2.3)

$$\mathbf{q} \cdot \nabla \theta = -\gamma \rho \theta^2 \leq 0$$

and $\gamma = \kappa |\nabla \theta|^2 / \rho \theta^2$ follows. Instead, in more involved models a further degree of generality is gained by letting (1) be an equation, not an identity.

The second fold is to point out that the CD inequality may result in a relation between appropriate rates and the entropy production. A representation formula allows the inequality to be solved with respect to a rate in terms of the remaining rates and of the entropy production. In this way we can establish the thermodynamic consistency of uncommon non-linear rate-type equations. Two models are developed in detail. First we examine a general thermodynamic scheme leading to the modelling of heat conduction and viscoelasticity. The application of the representation formula, for vectors and tensors, yields general relations; as particular cases some classical models of the literature are derived. Next, some models of heat conduction are investigated where the temperature rate is one of the variables. The possible models are framed within different schemes in the literature and the wave propagation properties are established.

We consider a body occupying a time-dependent region $\Omega \subset \mathscr{E}^3$. The motion is described by means of the function $\boldsymbol{\chi}(\mathbf{X}, t)$, providing the position vector $\mathbf{x} \in \Omega = \boldsymbol{\chi}(\mathrm{R}, t)$. The symbols $\nabla$ and $\nabla_R$ denote the gradient operator with respect to $\mathbf{x} \in \Omega$, $\mathbf{X} \in \mathrm{R}$. The function $\boldsymbol{\chi}$ is assumed to be differentiable; hence, we can define the deformation gradient as $\mathbf{F} = \nabla_R \boldsymbol{\chi}$ or, in suffix notation, $F_{iK} = \partial_{X_K} \chi_i$. The invertibility of $\mathbf{X} \to \mathbf{x} = \boldsymbol{\chi}(\mathbf{X}, t)$ is guaranteed by letting $J := \det \mathbf{F} > 0$. For any tensor $\mathbf{A}$ we define $|\mathbf{A}|$ as $(\mathbf{A} \cdot \mathbf{A})^{1/2}$. Throughout $(\mathbf{x}, t) \in \Omega \times \mathbb{R}$. We let $\mathbf{v}(\mathbf{x}, t)$ be the velocity field. For any function $f(\mathbf{x}, t)$ we let $\dot{f}$ be the total time derivative, $\dot{f} = \partial_t f + (\mathbf{v} \cdot \nabla) f$. A prime denotes the derivative of a function with respect to the argument.

## 2. Balance of Entropy and Statement of the Second Law

Let $\mathcal{P}_t$ be any sub-region of the body that is convected by the motion. As with any balance equation we may express the balance of entropy in $\mathcal{P}_t$ by letting the rate consist of a volume integral and a surface integral,

$$\frac{d}{dt} \int_{\mathcal{P}_t} \rho \eta \, dv = \int_{\mathcal{P}_t} \rho s \, dv - \int_{\partial \mathcal{P}_t} \mathbf{j} \cdot \mathbf{n} \, da,$$

where **j** is the entropy flux. Notice that if we start with a scalar integrand, say $h$, in the surface integral then a Cauchy-like theorem will lead to a linear dependence of $h$ on the unit normal **n**, say $h = -\mathbf{j} \cdot \mathbf{n}$. Within the physical scheme ([3], Chapter 6) the variation of entropy is greater than $\delta Q / \theta$, where $\delta Q$ is the heat transfer at the pertinent region while $\theta$

is the absolute temperature at that region. In the continuum setting we then let $s$ comprise $r/\theta$, $r$ being the energy supply, while $\mathbf{j}$ comprises $\mathbf{q}/\theta$, $\mathbf{q}$ being the heat flux. We then let

$$s = \frac{r}{\theta} + \gamma, \qquad \mathbf{j} = \frac{\mathbf{q}}{\theta} + \mathbf{k},$$

where $\mathbf{k}$ is the extra-entropy flux and $\gamma \geq 0$ is the entropy production (per unit mass). Consequently, we have

$$\int_{\mathcal{P}_t} [\rho \dot{\eta} - \frac{\rho r}{\theta} + \nabla \cdot (\frac{\mathbf{q}}{\theta} + \mathbf{k}) - \rho \gamma] dv = 0,$$

where, in local form,

$$\rho \dot{\eta} - \frac{\rho r}{\theta} + \nabla \cdot (\frac{\mathbf{q}}{\theta} + \mathbf{k}) = \rho \gamma \geq 0. \tag{2}$$

Equation (2) is the general form of the CD inequality. Following is the statement of the second law: *any thermodynamic process is required to satisfy the CD inequality (2)*.

For definiteness we consider a deformable solid. We then express the balance equations for mass, linear momentum, angular momentum and energy in the form

$$\dot{\rho} + \rho \nabla \cdot \mathbf{v} = 0, \tag{3}$$

$$\rho \dot{\mathbf{v}} = \rho \mathbf{b} + \nabla \cdot \mathbf{T}, \qquad \mathbf{T} = \mathbf{T}^T, \tag{4}$$

$$\rho \dot{\varepsilon} = \mathbf{T} \cdot \mathbf{D} + \rho r - \nabla \cdot \mathbf{q}, \tag{5}$$

where $\mathbf{b}$ is the body force and $\varepsilon$ is the internal energy density. Substitution of $\rho r - \nabla \cdot \mathbf{q}$ from (5) into (2) results in

$$\rho \theta \dot{\eta} - \rho \dot{\varepsilon} + \mathbf{T} \cdot \mathbf{D} + \theta \nabla \cdot \mathbf{k} - \frac{1}{\theta} \mathbf{q} \cdot \nabla \theta = \rho \theta \gamma.$$

Hence, by means of the Helmholtz free energy

$$\psi = \varepsilon - \theta \eta$$

we can write the inequality in the form

$$-\rho(\dot{\psi} + \eta \dot{\theta}) + \mathbf{T} \cdot \mathbf{D} + \theta \nabla \cdot \mathbf{k} - \frac{1}{\theta} \mathbf{q} \cdot \nabla \theta = \rho \theta \gamma. \tag{6}$$

In the application of the second law, and hence of inequality (6), we require that the thermodynamic process under consideration satisfies the balance Equations (3)–(5) with any functions $\mathbf{b}(\mathbf{x}, t)$ and $r(\mathbf{x}, t)$.

Rate-type equations are framed naturally in the Lagrangian description. In this connection quantities related to the reference configuration are denote by the index $R$. The referential mass density $\rho_R$, the second Piola stress $\mathbf{T}_{RR}$, and the referential vectors $\mathbf{q}_R, \mathbf{k}_R$ are defined by

$$\rho_R = \rho J, \qquad \mathbf{T}_{RR} = J \mathbf{F}^{-1} \mathbf{T} \mathbf{F}^{-T}, \qquad \mathbf{q}_R = J \mathbf{q} \mathbf{F}^{-T}, \qquad \mathbf{k}_R = J \mathbf{k} \mathbf{F}^{-T},$$

while

$$\dot{\mathbf{E}} = \mathbf{F}^T \mathbf{D} \mathbf{F}, \qquad \nabla \theta = \nabla_R \theta \mathbf{F}^{-1}, \qquad J \mathbf{q} \cdot \nabla \theta = \mathbf{q}_R \cdot \nabla_R \theta, \qquad J \nabla \cdot \mathbf{k} = \nabla_R \cdot \mathbf{k}_R.$$

Hence, the multiplication of (6) by $J$ yields

$$-\rho_R(\dot{\psi} + \eta \dot{\theta}) + \mathbf{T}_{RR} \cdot \dot{\mathbf{E}} + \theta \nabla_R \cdot \mathbf{k}_R - \frac{1}{\theta} \mathbf{q}_R \cdot \nabla_R \theta = \rho_R \theta \gamma. \tag{7}$$

For formal convenience hereafter we let $\psi_R = \rho_R \psi, \eta_R = \rho_R \eta$.

In the next section we investigate the restrictions placed by (6) and (7) in connection with generalized models for thermoelastic solids.

*Representation Formulae*

Assume we are given the equation

$$\mathbf{Z} \cdot \mathcal{K} + \mathbf{A} \cdot \mathcal{F} = f, \tag{8}$$

where $\mathbf{Z}, \mathcal{K}, \mathbf{A}, \mathcal{F}$ are second-order tensors and $f$ is a non-negative scalar. If $\mathcal{K}$ and $\mathcal{F}$ are arbitrary and independent then it follows that $\mathbf{Z} = \mathbf{0}, \mathbf{A} = \mathbf{0}$, and $f = 0$. Instead we suppose $\mathcal{K}$ and $\mathcal{F}$ are not independent and look for a relation between them.

Let $\mathbf{N}$ be a unit tensor, $|\mathbf{N}| = 1$. Then

$$\mathbf{Z} = (\mathbf{Z} \cdot \mathbf{N})\mathbf{N} + \mathbf{Z}_\perp, \qquad \mathbf{Z}_\perp \cdot \mathbf{N} = 0.$$

Assume $\mathbf{Z} \cdot \mathbf{N}$ is known, say $\mathbf{Z} \cdot \mathbf{N} = g$. Let $\otimes$ denote the dyadic product defined by

$$(\mathbf{A} \otimes \mathbf{B})\mathbf{C} = (\mathbf{B} \cdot \mathbf{C})\mathbf{A}$$

for any tensors $\mathbf{A}, \mathbf{B}, \mathbf{C}$. If $\mathbf{Z}_\perp$ is unknown then it may be expressed by

$$\mathbf{Z}_\perp = (\mathbf{I} - \mathbf{N} \otimes \mathbf{N})\mathbf{G} = \mathbf{G} - (\mathbf{G} \cdot \mathbf{N})\mathbf{N},$$

where $\mathbf{I}$ is the fourth-order unit tensor and $\mathbf{G}$ is an arbitrary second-order tensor. Hence, we can represent $\mathbf{Z}$ in the form

$$\mathbf{Z} = g\mathbf{N} + (\mathbf{I} - \mathbf{N} \otimes \mathbf{N})\mathbf{G}. \tag{9}$$

A strictly analogous representation formula holds for vectors, say $\mathbf{z}$, in the form

$$\mathbf{z} = g\mathbf{n} + (\mathbf{1} - \mathbf{n} \otimes \mathbf{n})\mathbf{w}, \tag{10}$$

where $\mathbf{w}$ is a vector, $\mathbf{n}$ a unit vector, and $\mathbf{1}$ the second-order unit tensor.

As an example, we return to (8) and let $\mathbf{N} = \mathcal{K}/|\mathcal{K}|$ so that

$$g = \mathbf{Z} \cdot \mathbf{N} = (f - \mathbf{A} \cdot \mathcal{F})/|\mathcal{K}|.$$

Hence, it follows from (9) that

$$\mathbf{Z} = \frac{f - \mathbf{A} \cdot \mathcal{F}}{|\mathcal{K}|^2}\mathcal{K} + (\mathbf{I} - \frac{\mathcal{K}}{|\mathcal{K}|} \otimes \frac{\mathcal{K}}{|\mathcal{K}|})\mathbf{G}.$$

## 3. Rate Equations and Euclidean Invariance

Rate-type models are often based on rheological analogues (e.g., [12], Chapter 8). Mathematically rate-type models are characterized by setting the time derivative of appropriate fields among the constitutive functions of the model. The interest in rate-type models is well motivated by a comment on hypo-elastic materials, described by

$$\dot{\mathbf{T}} = \hat{\mathbf{G}}(\mathbf{T}, \mathbf{L}),$$

versus materials with memory in that the entire kinematical history of a body can rarely be known ([13], § 99). The use of rate equations is standard in the extended irreversible thermodynamics [14,15]. Yet, as with any constitutive equation, the rate-type form is also required to comply with Euclidean invariance.

A change of frame $\mathscr{F} \to \mathscr{F}^*$ given by a Euclidean transformation, such that $\mathbf{x} \mapsto \mathbf{x}^*$, is expressed by

$$\mathbf{x}^* = \mathbf{c} + \mathbf{Q}\mathbf{x}, \qquad \mathbf{Q}^T\mathbf{Q} = \mathbf{1}. \tag{11}$$

Under the transformation (11), the deformation gradient $\mathbf{F}$ changes as a vector,

$$\mathbf{F}^* = \mathbf{Q}\mathbf{F},$$

and hence it is not invariant. Yet invariant scalars, vectors, and tensors occur in connection with $\mathbf{F}$.

We first look at invariants of mechanical character. The right Cauchy–Green tensor $\mathbf{C}$ and the Green–Lagrange (or Green–St. Venant) strain tensor $\mathbf{E}$, defined as

$$\mathbf{C} = \mathbf{F}^T\mathbf{F}, \qquad \mathbf{E} = \tfrac{1}{2}(\mathbf{C} - \mathbf{1}),$$

are invariant in that

$$\mathbf{C}^* = \mathbf{F}^{*T}\mathbf{F}^* = \mathbf{F}^T\mathbf{Q}^T\mathbf{Q}\mathbf{F} = \mathbf{F}^T\mathbf{F} = \mathbf{C}.$$

Consequently, the scalar

$$\mathbf{F} \cdot \mathbf{F} = \operatorname{tr}\mathbf{C} = 2\operatorname{tr}\mathbf{E} + 3$$

is invariant too. Since

$$\mathbf{L}^* = \mathbf{Q}\mathbf{L}\mathbf{Q}^T + \dot{\mathbf{Q}}\mathbf{Q}^T$$

then is apparently non-invariant. Decompose $\mathbf{L}$ in the classical form

$$\mathbf{L} = \mathbf{D} + \mathbf{W},$$

where $\mathbf{D}$ is the stretching tensor and $\mathbf{W}$ is the spin; we have

$$\mathbf{D}^* = \mathbf{Q}\mathbf{D}\mathbf{Q}^T, \qquad \mathbf{W}^* = \mathbf{Q}\mathbf{W}\mathbf{Q}^T + \dot{\mathbf{Q}}\mathbf{Q}^T.$$

The second Piola stress is invariant; this is checked by observing that

$$\mathbf{T}^*_{RR} = J(\mathbf{Q}\mathbf{F})^{-1}(\mathbf{Q}\mathbf{T}\mathbf{Q}^T)(\mathbf{Q}\mathbf{F})^{-T} = J\mathbf{F}^{-1}\mathbf{Q}^{-1}\mathbf{Q}\mathbf{T}\mathbf{Q}^T\mathbf{Q}^{-T}\mathbf{F}^{-T} = J\mathbf{F}^{-1}\mathbf{T}\mathbf{F}^{-T} = \mathbf{T}_{RR}.$$

We observe that since $\dot{\mathbf{E}} = \mathbf{F}^T\mathbf{D}\mathbf{F}$ then

$$\mathbf{T} \cdot \mathbf{D} = J^{-1}(\mathbf{F}\mathbf{T}_{RR}\mathbf{F}^T) \cdot \mathbf{D} = J^{-1}\mathbf{T}_{RR} \cdot (\mathbf{F}^T\mathbf{D}\mathbf{F}) = J^{-1}\mathbf{T}_{RR} \cdot \dot{\mathbf{E}}.$$

Hence, we have

$$\mathbf{T} \cdot \mathbf{D} = J^{-1}\mathbf{T}_{RR} \cdot \dot{\mathbf{E}}. \tag{12}$$

The referential heat flux and temperature gradient

$$\mathbf{q}_R = J\mathbf{F}^{-1}\mathbf{q}, \qquad \nabla_R\theta = \mathbf{F}^T\nabla\theta$$

are invariant and so is the power

$$\mathbf{q} \cdot \nabla\theta = J^{-1}\mathbf{q}_R \cdot \nabla_R\theta. \tag{13}$$

## 4. Thermodynamic Consistency of Thermo-Viscoelastic Solids

Here, we look for rate-type models of thermoelastic materials in that rate equations are considered for the stress tensor and the heat flux in deformable solids.

From the mechanical viewpoint we look for a scheme that accounts for a persistent rate of the response under a constant action (viscoelastic behaviour). For heat conduction the model is thought to describe both a non-instantaneous approach to stationarity and a higher-order spatial interaction. This suggests that we allow for rate equations of $\mathbf{T}$ and $\mathbf{q}$ and let $\dot{\theta}$ be a variable. Thus, we might take $(\theta, \mathbf{F}, \nabla\theta, \mathbf{T}, \mathbf{q}, \dot{\mathbf{F}}, \dot{\theta})$ as the set of independent variables. Yet, invariance requirements demand that the dependence on the derivatives

occurs in an objective way. Moreover, the Euclidean invariance of the free energy $\psi$ implies that the dependence of $\psi_R$ be through a function of Euclidean invariants. Hence, we let

$$\psi_R = \psi_R(\theta, \mathbf{E}, \mathbf{T}_{RR}, \mathbf{q}_R, \nabla_R\,\theta, \nabla_R\,\nabla_R\,\theta, \dot{\mathbf{E}}, \dot{\theta})$$

and the same for $\eta_R$ and $\gamma$. The constitutive assumptions are completed by letting the rates $\dot{\mathbf{T}}_{RR}$ and $\dot{\mathbf{q}}_R$ be given by constitutive functions of $\Gamma = (\theta, \mathbf{E}, \mathbf{T}_{RR}, \mathbf{q}_R, \nabla_R\,\theta, \nabla_R\,\nabla_R\,\theta, \dot{\mathbf{E}}, \dot{\theta})$.

Computing the time derivative of $\phi_R$ and substituting it into (7), one obtains

$$-(\partial_\theta\psi_R + \eta_R)\dot{\theta} + (\mathbf{T}_{RR} - \partial_{\mathbf{E}}\psi_R) \cdot \dot{\mathbf{E}} - \partial_{\mathbf{T}_{RR}}\psi_R \cdot \dot{\mathbf{T}}_{RR} - \partial_{\mathbf{q}_R}\psi_R \cdot \dot{\mathbf{q}}_R - \partial_{\nabla_R\,\theta}\psi_R \cdot \nabla_R\,\dot{\theta}$$

$$-\partial_{\nabla_R\,\nabla_R\,\theta}\psi_R \cdot \nabla_R\,\nabla_R\,\dot{\theta} - \partial_{\dot{\mathbf{E}}}\psi_R \cdot \ddot{\mathbf{E}} - \partial_{\dot{\theta}}\psi_R\ddot{\theta} - \frac{1}{\theta}\mathbf{q}_R \cdot \nabla_R\,\theta + \theta\nabla_R \cdot \mathbf{k}_R = \rho_R\theta\gamma, \quad (14)$$

where $\gamma \geq 0$. The (linearity and) arbitrariness of $\ddot{\theta}, \nabla_R\,\nabla_R\,\dot{\theta}$, and $\ddot{\mathbf{E}}$ implies that

$$\partial_{\dot{\theta}}\psi_R = \mathbf{0}, \qquad \partial_{\nabla_R\,\nabla_R\,\theta}\psi_R = \mathbf{0}, \qquad \partial_{\dot{\mathbf{E}}}\psi_R = \mathbf{0}.$$

Owing to the dependence of $\mathbf{k}_R$ on $\dot{\theta}$ it follows that

$$\nabla_R \cdot \mathbf{k}_R = \partial_{\dot{\theta}}\mathbf{k}_R \cdot \nabla_R\,\dot{\theta} + \ldots$$

where the dots denote possible terms which are independent of $\nabla_R\,\dot{\theta}$. Hence, the arbitrariness of $\nabla_R\,\dot{\theta}$ in (14) results in

$$-\partial_{\nabla_R\,\theta}\psi_R + \theta\partial_{\dot{\theta}}\mathbf{k}_R = \mathbf{0}.$$

Consequently,

$$\mathbf{k}_R = \frac{1}{\theta}\partial_{\nabla_R\,\theta}\psi_R\dot{\theta} + \hat{\mathbf{k}}(\theta, \mathbf{E}, \mathbf{T}_{RR}\mathbf{q}_R, \nabla_R\,\theta, \dot{\mathbf{E}}).$$

For the present purposes no significant generality is lost by letting $\hat{\mathbf{k}} = \mathbf{0}$. Thus, we have

$$\nabla_R \cdot \mathbf{k}_R = \frac{1}{\theta}\partial_{\nabla_R\,\theta}\psi_R \cdot \nabla_R\,\dot{\theta} + [\nabla_R \cdot (\frac{1}{\theta}\partial_{\nabla_R\,\theta}\psi_R)]\dot{\theta}.$$

Substitution into (14) yields

$$-(\delta_\theta\psi_R + \eta_R)\dot{\theta} + (\mathbf{T}_{RR} - \partial_{\mathbf{E}}\psi_R) \cdot \dot{\mathbf{E}} - \partial_{\mathbf{T}_{RR}}\psi_R \cdot \dot{\mathbf{T}}_{RR} - \partial_{\mathbf{q}_R}\psi_R \cdot \dot{\mathbf{q}}_R - \frac{1}{\theta}\mathbf{q}_R \cdot \nabla_R\,\theta = \rho_R\theta\gamma, \quad (15)$$

where

$$\delta_\theta\psi_R := \partial_\theta\psi_R - \theta\,\nabla_R \cdot (\frac{1}{\theta}\partial_{\nabla_R\,\theta}\,\psi_R).$$

Notice that, since $\psi_R$ depends on $\nabla_R\,\theta$, then $\delta_\theta\psi_R$ may depend on $\nabla_R\,\nabla_R\,\theta$. Only $\gamma$ and $\eta$ can depend on $\dot{\theta}$.

The unknown functions $\dot{\mathbf{T}}_{RR}, \dot{\mathbf{q}}_R$, and $\eta_R$ can be related by common dependencies so that cross-coupling terms are allowed. For simplicity we examine a sufficient condition for the validity of (15) namely that the three inequalities

$$-(\delta_\theta\psi_R + \eta_R)\dot{\theta} = \rho_R\theta\gamma_\theta \geq 0, \quad (16)$$

$$(\mathbf{T}_{RR} - \partial_{\mathbf{E}}\psi_R) \cdot \dot{\mathbf{E}} - \partial_{\mathbf{T}_{RR}}\psi_R \cdot \dot{\mathbf{T}}_{RR} = \rho_R\theta\gamma_T \geq 0, \quad (17)$$

$$-\partial_{\mathbf{q}_R}\psi_R \cdot \dot{\mathbf{q}}_R - \frac{1}{\theta}\mathbf{q}_R \cdot \nabla_R\,\theta = \rho_R\theta\gamma_q \geq 0 \quad (18)$$

are satisfied while $\gamma = \gamma_\theta + \gamma_T + \gamma_q$. For definiteness we now let

$$\psi_R(\theta, \mathbf{E}, \mathbf{T}_{RR}, \mathbf{q}_R, \nabla_R\,\theta) = \psi_T(\theta, \mathbf{E}, \mathbf{T}_{RR}) + \psi_q(\theta, \mathbf{q}_R, \nabla_R\,\theta).$$

## 4.1. Consequences of (18); Heat Conduction

As for Equation (18), we observe that if $\partial_{\mathbf{q}_R} \psi_R = \mathbf{0}$ then it follows the heat equation

$$-\frac{1}{\theta}\mathbf{q}_R \cdot \nabla_R \theta = \rho_R \theta \gamma_q \geq 0$$

which is satisfied by any function

$$\mathbf{q}_R = -\kappa(\theta, |\nabla_R \theta|)\mathbf{K}\nabla_R \theta$$

where $\kappa > 0$ and $\mathbf{K}$ is positive definite and hence

$$\rho_R \theta \gamma_q = \kappa \nabla_R \theta \cdot \mathbf{K}\nabla_R \theta.$$

As a particular example, let $\mathbf{K} = J^{-1}\mathbf{B}^{-1}$. Then

$$\mathbf{q}_R = -\kappa J^{-1}\mathbf{F}^{-1}\mathbf{F}^{-T}\nabla_R \theta = -\kappa J^{-1}\mathbf{F}^{-1}\nabla\theta, \qquad \mathbf{q} = -\kappa \nabla\theta,$$

which is Fourier's law.

If instead $\partial_{\mathbf{q}_R} \psi_R \neq \mathbf{0}$ then we can apply (10), with $\mathbf{z} = \dot{\mathbf{q}}_R$ and $\mathbf{n} = \partial_{\mathbf{q}_R} \psi_R / |\partial_{\mathbf{q}_R} \psi_R|$, to obtain

$$\dot{\mathbf{q}}_R = \frac{\dot{\mathbf{q}}_R \cdot \partial_{\mathbf{q}_R} \psi_R}{|\partial_{\mathbf{q}_R} \psi_R|^2}\partial_{\mathbf{q}_R} \psi_R + \left(\mathbf{1} - \frac{\partial_{\mathbf{q}_R} \psi_R \otimes \partial_{\mathbf{q}_R} \psi_R}{|\partial_{\mathbf{q}_R} \psi_R|^2}\right)\mathbf{w}$$

$$= -\frac{\theta^{-1}\mathbf{q}_R \cdot \nabla_R \theta + \rho_R \theta \gamma_q}{|\partial_{\mathbf{q}_R} \psi_R|^2}\partial_{\mathbf{q}_R} \psi_R + \left(\mathbf{1} - \frac{\partial_{\mathbf{q}_R} \psi_R \otimes \partial_{\mathbf{q}_R} \psi_R}{|\partial_{\mathbf{q}_R} \psi_R|^2}\right)\mathbf{w}.$$

As an example, let

$$\partial_{\mathbf{q}_R} \psi_R = \alpha(\theta)\mathbf{q}_R, \qquad \mathbf{w} = \beta(\theta)\nabla_R \theta.$$

Hence, we have

$$\dot{\mathbf{q}}_R = -\frac{\rho_R \theta \gamma_q}{\alpha|\mathbf{q}_R|^2}\mathbf{q}_R - \frac{\mathbf{q}_R \cdot \nabla_R \theta}{\theta\alpha|\mathbf{q}_R|^2}\mathbf{q}_R + \beta\nabla_R \theta - \beta\frac{\mathbf{q}_R \cdot \nabla_R \theta}{|\mathbf{q}_R|^2}\mathbf{q}_R.$$

The choice $\beta = -1/\theta\alpha$ results in

$$\dot{\mathbf{q}}_R + \frac{\rho_R \theta \gamma_q}{\alpha|\mathbf{q}_R|^2}\mathbf{q}_R = -\frac{1}{\theta\alpha}\nabla_R \theta. \tag{19}$$

Equation (19) has the form of a Maxwell–Cattaneo (MC for short) Equation (see, e.g., [16]) with relaxation time $\tau$ and conductivity $\kappa$ given by

$$\tau = \frac{\alpha|\mathbf{q}_R|^2}{\rho_R \theta \gamma_q}, \qquad \kappa = \frac{|\mathbf{q}_R|^2}{\rho_R \gamma_q \theta^2}.$$

As expected, the positiveness of $\gamma_q$ results in the positiveness of the conductivity $\kappa$.

## 4.2. Consequences of (16)

Let

$$\hat{\eta} = \eta_R(\Gamma) - \eta_0, \qquad \eta_0 = \eta_R(\theta, \mathbf{E}, \mathbf{T}_{RR}, \mathbf{q}_R, \nabla_R \theta, \nabla_R \nabla_R \theta, \dot{\mathbf{E}}, 0).$$

The continuity of $\eta_R$ implies that $\hat{\eta} \to 0$ as $\dot{\theta} \to 0$. Consequently, for sufficiently small $|\dot{\theta}|$ we have

$$\mathrm{sgn}\gamma_\theta = -\mathrm{sgn}(\delta_\theta \psi_R + \eta_0)\dot{\theta}.$$

The arbitrary sign of $\dot{\theta}$ implies

$$\delta_\theta \psi_R + \eta_0 = 0$$

and hence

$$\hat{\eta}(\Gamma)\dot{\theta} = -\rho_R \theta \gamma_\theta \leq 0.$$

As an example, we may have

$$\hat{\eta} = \alpha(\theta, \mathbf{E})\dot{\theta}, \qquad \rho_R \theta \gamma_\theta = -\alpha \dot{\theta}^2, \qquad \alpha < 0.$$

The dependence on $\dot{\theta}$ has also been considered in an attempt to establish a model allowing for wave propagation at finite speed. For simplicity, assume the body is undeformable ($\mathbf{E} = \mathbf{0}$) and $\partial_{\nabla\theta}\psi = \mathbf{0}$. The evolution of $\theta$ is governed by the balance of energy,

$$\rho\dot{\varepsilon} = -\nabla \cdot \mathbf{q} + \rho r. \tag{20}$$

Since

$$\psi = \psi(\theta), \qquad \eta = -\psi' + \alpha\dot{\theta}$$

then

$$\rho\dot{\varepsilon} = \alpha\ddot{\theta} + \alpha\dot{\theta}^2 + \theta\eta_0'\dot{\theta}.$$

If $\mathbf{q} = -\kappa\nabla\theta$, $\kappa > 0$ then (20) yields

$$\rho\alpha\ddot{\theta} + \rho(\alpha\dot{\theta}^2 + \theta\eta_0'\dot{\theta}) - \kappa\Delta\theta = \rho r. \tag{21}$$

Since $\alpha < 0$ then (21) is an elliptic differential equation, not a hyperbolic one.

Furthermore, the more involved model with the dependence on $\nabla\theta$ would not result in a hyperbolic equation.

Though it is unusual in the literature, we might consider constitutive equations where both $\theta$ and $\eta_R$ are in the set of variables. In this case, inequality (16) would be in the form

$$-(\delta_\theta \psi_R + \eta_R)\dot{\theta} - \partial_{\eta_R}\psi_R \dot{\eta}_R = \rho_R \theta \gamma_\theta \geq 0,$$

thus resulting in the rate equation

$$\dot{\eta}_R = -\frac{\delta_\theta \psi_R + \eta_R}{\partial_{\eta_R}\psi_R}\dot{\theta} - \frac{\rho_R \theta \gamma_\theta}{\partial_{\eta_R}\psi_R}. \tag{22}$$

To establish the physical relevance of Equation (22), a careful investigation is required.

### 4.3. Consequences of (17); Viscoelasticity

If $\dot{\mathbf{E}}$ and $\dot{\mathbf{T}}_{RR}$ are independent of each other then we have

$$\partial_{\mathbf{T}_{RR}}\psi_T = \mathbf{0}, \qquad \mathbf{T}_{RR} = \partial_{\mathbf{E}}\psi_T,$$

and $\gamma_T = 0$. Hence, $\psi_T$ depends only on $\theta, \mathbf{E}$. Furthermore, $\mathbf{T}_{RR}$ is no longer an independent variable but is equal to $\partial_{\mathbf{E}}\psi_T$. This relation can be viewed as a model of a thermo-hyperelastic material.

We now consider $\gamma_T(\theta, \mathbf{E}, \mathbf{T}_{RR}, \dot{\mathbf{E}})$ and $\psi_T(\theta, \mathbf{E}, \mathbf{T}_{RR})$, with the assumption $\partial_{\mathbf{T}_{RR}}\psi_T \neq \mathbf{0}$. Equation (17), i.e.,

$$\partial_{\mathbf{T}_{RR}}\psi_T \cdot \dot{\mathbf{T}}_{RR} = (\mathbf{T}_{RR} - \partial_{\mathbf{E}}\psi_R) \cdot \dot{\mathbf{E}} - \rho_R \theta \gamma_T,$$

yields $\partial_{\mathbf{T}_{RR}}\psi_T \cdot \dot{\mathbf{T}}_{RR}$ as a function of $\theta, \mathbf{E}, \dot{\mathbf{E}}$. Using the representative Formula (9), with $\mathbf{Z} = \dot{\mathbf{T}}_{RR}$ we find

$$\dot{\mathbf{T}}_{RR} = \frac{(\mathbf{T}_{RR} - \partial_{\mathbf{E}}\psi_T) \cdot \dot{\mathbf{E}} - \rho_R \theta \gamma_T}{|\partial_{\mathbf{T}_{RR}}\psi_T|^2}\partial_{\mathbf{T}_{RR}}\psi_T + (\mathbf{I} - \frac{\partial_{\mathbf{T}_{RR}}\psi_T \otimes \partial_{\mathbf{T}_{RR}}\psi_T}{|\partial_{\mathbf{T}_{RR}}\psi_T|^2})\mathbf{G} \tag{23}$$

where $\mathbf{G}$ is any second-order tensor function of $\theta, \mathbf{E}, \mathbf{T}_{RR}, \dot{\mathbf{E}}$. Among the possible forms of $\mathbf{G}$, we consider

$$\mathbf{G} = \mathbf{J}_{RR}(\theta, \mathbf{E}, \mathbf{T}_{RR})\dot{\mathbf{E}},$$

where $\mathbf{J}_{RR}$ is a fourth-order tensor, which maximizes the (linear) dependence on $\dot{\mathbf{E}}$. Hence, it follows

$$\dot{\mathbf{T}}_{RR} = \mathbf{C}_{RR}(\theta, \mathbf{E}, \mathbf{T}_{RR})\dot{\mathbf{E}} - \frac{\rho_R \theta \gamma_T}{|\partial_{\mathbf{T}_{RR}}\psi_T|^2}\partial_{\mathbf{T}_{RR}}\psi_T, \tag{24}$$

where

$$\mathbf{C}_{RR} = \mathbf{J}_{RR} + \frac{\partial_{\mathbf{T}_{RR}}\psi_T \otimes (\mathbf{T}_{RR} - \partial_{\mathbf{E}}\psi_T - \mathbf{J}_{RR}^T\partial_{\mathbf{T}_{RR}}\psi_T}{|\partial_{\mathbf{T}_{RR}}\psi_T|^2}.$$

The result (24) gives a general representation of viscoelastic modelling.

Of course, the entropy production $\gamma_T$ may depend on $\dot{\mathbf{E}}$. Indeed, $\gamma_T$ has to be non-negative and a dependence of $\gamma_T$ on $|\dot{\mathbf{E}}|$ has been shown to be the natural modelling of the hysteretic behaviour in plastic materials [9].

The constitutive relation (23) has the form

$$\dot{\mathbf{T}}_{RR} = \boldsymbol{\mathcal{T}}(\theta, \mathbf{E}, \mathbf{T}_{RR}, \dot{\mathbf{E}})$$

and is linear in $\dot{\mathbf{E}}$ if $\mathbf{G} = \mathbf{0}$ and $\gamma_T = 0$. This equation can be viewed as characterizing hypo-elastic materials in that they experience a stress increase arising in response to the rate of strain $\dot{\mathbf{E}}$ from the immediately preceding state ([1], page 731). In fact, Truesdell [17] restricted hypo-elasticity to $\dot{\mathbf{T}} = \mathbf{DL}$ with $\mathbf{D}$ depending on the stress; in elasticity $\mathbf{D}$ also depends on the strain.

We now return to (24) and look for a simple example of $\mathbf{C}_{RR}$ induced by the free energy $\psi_T$, while $\mathbf{J}_{RR} = \mathbf{0}$ and $\gamma_T = 0$. Let $\mathbf{M}$ be a non-singular, fully-symmetric, fourth-order tensor and $\boldsymbol{\mathcal{G}}$ a smooth function from Sym to Sym. Hence, we consider the free energy

$$\psi_T = \psi_0(\theta) + \int_0^{\mathbf{E}} \boldsymbol{\mathcal{G}}(\boldsymbol{\mathcal{E}}) \cdot d\boldsymbol{\mathcal{E}} + \tfrac{1}{2}[\mathbf{T}_{RR} - \boldsymbol{\mathcal{G}}(\mathbf{E})] \cdot \mathbf{M}(\theta)[\mathbf{T}_{RR} - \boldsymbol{\mathcal{G}}(\mathbf{E})], \tag{25}$$

where $\boldsymbol{\mathcal{E}} \in$ Sym. Substitution of

$$\partial_{\mathbf{T}_{RR}}\psi_T = \mathbf{M}(\theta)[\mathbf{T}_{RR} - \boldsymbol{\mathcal{G}}(\mathbf{E})], \qquad \partial_{\mathbf{E}}\psi_T = \boldsymbol{\mathcal{G}}(\mathbf{E}) - [\boldsymbol{\mathcal{G}}'(\mathbf{E})]^T \mathbf{M}(\theta)[\mathbf{T}_{RR} - \boldsymbol{\mathcal{G}}(\mathbf{E})]$$

results in

$$\dot{\mathbf{T}}_{RR} = [\boldsymbol{\mathcal{G}}'(\mathbf{E}) + \mathbf{M}^{-1}(\theta)]\dot{\mathbf{E}}.$$

In the linear case, $\boldsymbol{\mathcal{G}}(\mathbf{E}) = \mathbf{LE}$, with $\mathbf{L}$ being a fourth-order tensor, we have

$$\dot{\mathbf{T}}_{RR} = \mathbf{C}_{RR}\dot{\mathbf{E}}, \qquad \mathbf{C}_{RR} := \mathbf{L} + \mathbf{M}^{-1}.$$

Now we show that the general form (23) of the representation of $\dot{\mathbf{T}}_{RR}$ also allows us to find very simple models of viscoelasticity. Consider again the free energy (25) and look for an entropy production given by

$$\rho_R \theta \gamma_T = \frac{1}{\tau}[\mathbf{T}_{RR} - \boldsymbol{\mathcal{G}}(\mathbf{E})] \cdot \mathbf{M}[\mathbf{T}_{RR} - \boldsymbol{\mathcal{G}}(\mathbf{E})]; \tag{26}$$

to save writing we let $\boldsymbol{\mathcal{G}}$ stand for $\boldsymbol{\mathcal{G}}(\mathbf{E})$. Notice that

$$(\mathbf{T}_{RR} - \partial_{\mathbf{E}}\psi_T) = \mathbf{T}_{RR} - \boldsymbol{\mathcal{G}} + \boldsymbol{\mathcal{G}}'^T \mathbf{M}(\mathbf{T}_{RR} - \boldsymbol{\mathcal{G}}) = (\mathbf{M}^{-1} + \boldsymbol{\mathcal{G}}'^T)\mathbf{M}[\mathbf{T}_{RR} - \boldsymbol{\mathcal{G}}].$$

Hence, since $\partial_{\mathbf{T}_{RR}} \psi_T = \mathbf{M}(\mathbf{T}_{RR} - \boldsymbol{\mathcal{G}})$ we obtain

$$\frac{(\mathbf{T}_{RR} - \partial_{\mathbf{E}}\psi_T) \cdot \dot{\mathbf{E}}}{|\partial_{\mathbf{T}_{RR}}\psi_T|^2} \partial_{\mathbf{T}_{RR}}\psi_T = [(\mathbf{M}^{-1} + \boldsymbol{\mathcal{G}}'^T(\mathbf{E}))\mathbf{M}(\mathbf{T}_{RR} - \boldsymbol{\mathcal{G}})] \cdot \dot{\mathbf{E}}|\mathbf{M}(\mathbf{T}_{RR} - \boldsymbol{\mathcal{G}})|^2 \mathbf{M}(\mathbf{T}_{RR} - \boldsymbol{\mathcal{G}})$$

$$= \frac{\mathbf{M}(\mathbf{T}_{RR} - \boldsymbol{\mathcal{G}}) \otimes \mathbf{M}(\mathbf{T}_{RR} - \boldsymbol{\mathcal{G}})}{|\mathbf{M}(\mathbf{T}_{RR} - \boldsymbol{\mathcal{G}})|^2}(\mathbf{M}^{-1} + \boldsymbol{\mathcal{G}}')\dot{\mathbf{E}} = (\mathbf{N} \otimes \mathbf{N})[(\mathbf{M}^{-1} + \boldsymbol{\mathcal{G}}')\dot{\mathbf{E}}],$$

where $\mathbf{N} = \mathbf{M}(\mathbf{T}_{RR} - \boldsymbol{\mathcal{G}})/|\mathbf{M}(\mathbf{T}_{RR} - \boldsymbol{\mathcal{G}})|$. Further, by (26) and (25) we find

$$\frac{\rho_R \theta \gamma_T}{|\partial_{\mathbf{T}_{RR}}\psi_T|^2} \partial_{\mathbf{T}_{RR}}\psi_T = \frac{1}{\tau}(\mathbf{N} \otimes \mathbf{N})(\mathbf{T}_{RR} - \boldsymbol{\mathcal{G}}).$$

Substitution into (23) yields

$$\dot{\mathbf{T}}_{RR} = (\mathbf{N} \otimes \mathbf{N})[(\mathbf{M}^{-1} + \boldsymbol{\mathcal{G}}')\dot{\mathbf{E}} - \frac{1}{\tau}(\mathbf{T}_{RR} - \boldsymbol{\mathcal{G}})] + (\mathbf{I} - \mathbf{N} \otimes \mathbf{N})\mathbf{G}. \tag{27}$$

Whenever the rate equation has the form

$$\dot{\mathbf{T}}_{RR} = (\mathbf{N} \otimes \mathbf{N})\hat{\mathbf{T}} + (\mathbf{I} - \mathbf{N} \otimes \mathbf{N})\mathbf{G},$$

for any tensor $\hat{\mathbf{T}}$ the choice $\mathbf{G} = \hat{\mathbf{T}}$ results in

$$\dot{\mathbf{T}}_{RR} = \hat{\mathbf{T}}.$$

Consequently, letting $\mathbf{G} = [(\mathbf{M}^{-1} + \boldsymbol{\mathcal{G}}')\dot{\mathbf{E}} - \tau^{-1}(\mathbf{T}_{RR} - \boldsymbol{\mathcal{G}})]$ we have

$$\dot{\mathbf{T}}_{RR} = [(\mathbf{M}^{-1} + \boldsymbol{\mathcal{G}}')\dot{\mathbf{E}} - \frac{1}{\tau}(\mathbf{T}_{RR} - \boldsymbol{\mathcal{G}})]. \tag{28}$$

For definiteness, let $\boldsymbol{\mathcal{G}}$ be linear, namely $\boldsymbol{\mathcal{G}} = \mathbf{G}_\infty \mathbf{E}$, and define $\mathbf{G}_0 = \mathbf{G}_\infty + \mathbf{M}^{-1}$ [18]. Hence, Equation (28) becomes

$$\dot{\mathbf{T}}_{RR} + \frac{1}{\tau}(\mathbf{T}_{RR} - \mathbf{G}_\infty \mathbf{E}) = \mathbf{G}_0 \dot{\mathbf{E}},$$

which may be viewed as a three-dimensional version of the standard linear solid.

## 5. A Rate-Type Approach to the Equation for the Heat Flux

Equation (18) is an implicit relation for the heat flux $\mathbf{q}_R$. This is consistent with the fact that Equation (19) is derived by assuming $\partial_{\mathbf{q}_R}\psi_R \neq \mathbf{0}$. Hence, if $\partial_{\mathbf{q}_R}\psi_R = \alpha\mathbf{q}_R$, the consistency follows by requiring that $\alpha \neq 0, \mathbf{q}_R \neq \mathbf{0}$. This might appear as a restriction on the consistency of the MC Equation (19). Furthermore, in the investigation of wave propagation properties the requirement $\mathbf{q}_R \neq \mathbf{0}$ implies that we cannot account for discontinuity waves propagating in a region with $\mathbf{q}_R = \mathbf{0}$. Hence, we re-examine the consistency of heat-flux equations by starting with a rate-type form of the sought equation. For simplicity we let the body be rigid and at rest in the chosen frame of reference. Consequently, $\mathbf{q}_R = \mathbf{q}, \nabla_R \theta = \nabla\theta, \mathbf{k}_R = \mathbf{k}$. The CD inequality then simplifies to

$$-\rho(\dot{\psi} + \eta\dot{\theta}) - \frac{1}{\theta}\mathbf{q} \cdot \nabla\theta + \theta\nabla \cdot \mathbf{k} = \rho\theta\gamma_q. \tag{29}$$

Let $\theta, \mathbf{q}$ and $\nabla\theta$ be the variables. Hence, $\psi, \eta, \gamma_q$ and $\mathbf{k}$ are given by functions of $\theta, \mathbf{q}, \nabla\theta$ and $\dot{\mathbf{q}}$ is taken in the form

$$\dot{\mathbf{q}} = -f\,\mathbf{q} - g\,\nabla\theta,$$

subject to $f \geq 0$. The scalars $\psi, \eta, \gamma_q, f$, and $g$ are assumed to depend on $\mathbf{q}$ and $\nabla\theta$ through the magnitudes $q = |\mathbf{q}|$ and $l = |\nabla\theta|$. Computation of $\dot{\psi}$ and substitution in (29) yield

$$-\rho(\partial_\theta\psi + \eta)\dot{\theta} - \rho\partial_{\nabla\theta}\psi \cdot \nabla\dot{\theta} + \rho f \partial_\mathbf{q}\psi \cdot \mathbf{q} + (\rho g \partial_\mathbf{q}\psi - \frac{1}{\theta}\mathbf{q}) \cdot \nabla\theta + \theta\nabla \cdot \mathbf{k} = \rho\theta\gamma_q.$$

The arbitrariness of $\nabla\dot{\theta}$ and $\dot{\theta}$ imply that

$$\partial_{\nabla\theta}\psi = \mathbf{0}, \qquad \eta = -\partial_\theta\psi.$$

Likewise, we find that $\mathbf{k}$ might only depend on $\theta$; we loose no generality by letting $\mathbf{k} = \mathbf{0}$. The remaining inequality is

$$\rho f \partial_\mathbf{q}\psi \cdot \mathbf{q} + (\rho g \partial_\mathbf{q}\psi - \frac{1}{\theta}\mathbf{q}) \cdot \nabla\theta = \rho\theta\gamma_q.$$

Since $\mathbf{q}$ and $\nabla\theta$ are independent vectors then the arbitrariness of $\nabla\theta$ implies

$$\rho g \partial_\mathbf{q}\psi = \frac{1}{\theta}\mathbf{q}, \qquad \rho f \partial_\mathbf{q}\psi \cdot \mathbf{q} = \rho\theta\gamma_q;$$

it then follows that $g$ is independent of $\nabla\theta$. Hence, we find the entropy production in the form

$$\gamma_q = \frac{f}{\rho g \theta^2} q^2, \qquad g > 0.$$

Furthermore, the dependence of $\psi$ on $\mathbf{q}$ through $q$ leads to

$$\psi(\theta, q) = \frac{1}{\rho\theta} \int_0^q \frac{\xi}{g(\theta, \xi)} d\xi + \psi_0(\theta).$$

An analogous model for anisotropic solids is obtained by letting

$$\dot{\mathbf{q}} = -\mathbf{\Lambda}\mathbf{q} - \mathbf{K}\nabla\theta, \tag{30}$$

where $\mathbf{K}(\theta) \in \text{Sym}$ while $\mathbf{\Lambda}(\theta)$ is a positive definite. Hence, $\mathbf{\Lambda}^{-1}\mathbf{K}$ plays the role of a conductivity tensor under stationary conditions. We assume that $\mathbf{K}$ and $\mathbf{\Lambda}$ have a common basis of eigenvectors. By paralleling the previous derivation it follows from (29) that

$$\partial_\mathbf{q}\psi\mathbf{K} = \frac{1}{\rho\theta}\mathbf{q}, \qquad \theta\gamma_q = \partial_\mathbf{q}\psi \cdot \mathbf{\Lambda}\mathbf{q}.$$

Consequently, we have

$$\psi(\theta, \mathbf{q}) = \frac{1}{2\rho\theta}\mathbf{q} \cdot \mathbf{K}^{-1}\mathbf{q} + \psi_0(\theta),$$

$$\gamma_q = \frac{1}{\rho\theta^2}\mathbf{q} \cdot (\mathbf{K}^{-1}\mathbf{\Lambda})\mathbf{q}.$$

Thus, $\mathbf{K}^{-1}\mathbf{\Lambda} > \mathbf{0}$ and this in turn implies the positive definiteness of the conductivity tensor $\mathbf{\Lambda}^{-1}\mathbf{K}$. Since $\mathbf{\Lambda}^{-1} > \mathbf{0}$ then $\mathbf{K} > \mathbf{0}$.

We now check the wave properties associated with (30) and the balance of energy for solids,

$$\rho\dot{\varepsilon} = -\nabla \cdot \mathbf{q} + \rho r. \tag{31}$$

We consider discontinuity waves and denote the difference between the limit value behind ($w_-$) and ahead ($w_+$) of the wave by $[\![w]\!] = w_- - w_+$. We investigate weak discontinuities in that we assume the jump conditions

$$[\![\theta]\!] = 0, \qquad [\![\mathbf{q}]\!] = \mathbf{0}, \qquad [\![r]\!] = 0.$$

Let $U$ be the wave speed. By means of geometrical and kinematical conditions ([1], § 172) we have

$$[\![\dot{\mathbf{q}}]\!] = -U[\![\partial_n \mathbf{q}]\!], \quad [\![\nabla \cdot \mathbf{q}]\!] = \mathbf{n} \cdot [\![\partial_n \mathbf{q}]\!], \quad [\![\dot{\theta}]\!] = -U[\![\partial_n \theta]\!], \quad [\![\nabla \theta]\!] = [\![\partial_n \theta]\!]\mathbf{n},$$

where $\mathbf{n}$ is the unit normal to the wave and $\partial_n$ denotes the normal derivative. Notice that

$$\rho\dot{\varepsilon} = \beta\dot{\theta} + \mathbf{p} \cdot \dot{\mathbf{q}}, \qquad \beta := \rho\theta\partial_\theta\eta, \qquad \mathbf{p} := \rho(\partial_{\mathbf{q}}\psi - \theta\partial_{\mathbf{q}}\partial_\theta\psi);$$

where $\beta > 0$ is the specific heat and $\mathbf{p} = \mathbf{0}$ if $\mathbf{q} = \mathbf{0}$. By (30) it follows

$$U[\![\partial_n \mathbf{q}]\!] = \mathbf{Kn}[\![\partial_n \theta]\!].$$

Hence, by (30) and (31), and some algebraic manipulations we obtain

$$\left(\beta U^2 + \mathbf{p} \cdot \mathbf{Kn}\, U - \mathbf{n} \cdot \mathbf{Kn}\right)[\![\partial_n \theta]\!] = 0.$$

Non-zero discontinuities can propagate with speeds

$$U_\pm = -\frac{\mathbf{p} \cdot \mathbf{Kn}}{2\beta} \pm \Big[\frac{\mathbf{n} \cdot \mathbf{Kn}}{\beta} + \Big(\frac{\mathbf{p} \cdot \mathbf{Kn}}{2\beta}\Big)^2\Big]^{1/2}.$$

In particular, waves entering a region where $\mathbf{q} = \mathbf{0}$ (and hence $\mathbf{p} = \mathbf{0}$) propagate with the speed $(\mathbf{n} \cdot \mathbf{Kn}/\beta)^{1/2}$. Hence, greater tensor $\mathbf{K}$ increases the speed.

## 6. Models Involving the Temperature Rate

Section 4 shows the difficulties of modelling heat conduction with finite speed in terms of the dependence on $\dot{\theta}$ for rigid bodies. Here, we look for a more general model where the dependence on $\dot{\theta}$ is considered for thermo-elastic bodies. Due to deformation, the model is simpler within the reference configuration.

Our purpose is to let $\theta, \mathbf{E}, \mathbf{q}_R, \nabla_R \theta$ and $\dot{\theta}$ be independent variables. Yet, the occurrence of $\nabla_R \theta, \dot{\theta}$ and the equipresence principle also show that the dependence on $\nabla_R \nabla_R \theta$ is in order. Hence, we let $\psi_R, \eta_R, \mathbf{T}_{RR}, \gamma$ depend on

$$\Gamma = (\theta, \mathbf{E}, \mathbf{q}_R, \nabla_R \theta, \dot{\theta}, \nabla_R \nabla_R \theta)$$

and, by analogy with Section 5, we consider the equation

$$\dot{\mathbf{q}}_R = -f(\theta, \mathbf{E})\mathbf{q}_R - \mathbf{K}_R(\theta, \mathbf{E})\nabla_R \theta, \qquad \mathbf{K}_R \in \text{Sym},$$

for the evolution of $\mathbf{q}_R$. Hence, the referential CD inequality (7) takes the form

$$-(\partial_\theta \psi_R + \eta_R)\dot{\theta} + (\mathbf{T}_{RR} - \partial_{\mathbf{E}}\psi_R) \cdot \dot{\mathbf{E}} - \partial_{\nabla_R \theta}\psi_R \cdot \nabla_R \dot{\theta} - \partial_{\nabla_R \nabla_R \theta}\psi_R \cdot \nabla_R \nabla_R \dot{\theta} - \partial_{\dot{\theta}}\psi_R \ddot{\theta}$$
$$+ f\partial_{\mathbf{q}_R}\psi_R \cdot \mathbf{q}_R - \Big(\frac{1}{\theta}\mathbf{q}_R + \partial_{\mathbf{q}_R}\psi_R \mathbf{K}_R\Big) \cdot \nabla_R \theta + \theta\nabla_R \cdot \mathbf{k}_R = \rho_R\theta\gamma \qquad (32)$$

It follows that

$$\partial_{\dot{\theta}}\psi_R = 0, \qquad \partial_{\nabla_R \nabla_R \theta}\psi_R = \mathbf{0}, \qquad \partial_{\nabla_R \theta}\psi_R = \theta\partial_{\dot{\theta}}\mathbf{k}_R.$$

Hence, with no significant loss of generality, we let

$$\mathbf{k}_R = \frac{1}{\theta}\partial_{\nabla_R \theta}\psi_R \dot{\theta}.$$

The arbitrariness of $\dot{\mathbf{E}}$ implies

$$\mathbf{T}_{RR} = \partial_{\mathbf{E}}\psi_R.$$

Hence, the inequality (32) simplifies to

$$-(\delta_\theta \psi_R + \eta_R)\dot{\theta} + f \partial_{\mathbf{q}_R} \psi_R \cdot \mathbf{q}_R - (\frac{1}{\theta}\mathbf{q}_R + \partial_{\mathbf{q}_R}\psi_R \mathbf{K}_R) \cdot \nabla_R \theta = \rho_R \theta \gamma.$$

Since $\psi_R$ is independent of $\dot{\theta}$ then it follows

$$-(\delta_\theta \psi_R + \eta_R)\dot{\theta} \geq 0.$$

Letting

$$\eta_R = \eta_0(\theta, \mathbf{E}, \mathbf{q}_R, \nabla_R \theta, \nabla_R \nabla_R \theta) + \hat{\eta}(\Gamma), \qquad \hat{\eta} \to 0 \text{ as } \dot{\theta} \to 0$$

we find that

$$\eta_0 = -\delta_\theta \psi_R, \tag{33}$$

$$\hat{\eta}\,\dot{\theta} \leq 0, \quad \hat{\eta} = -\alpha(\Gamma)\dot{\theta}, \quad \alpha \geq 0. \tag{34}$$

If instead we let $\dot{\theta} = 0$ then we have

$$f \partial_{\mathbf{q}_R} \psi_R \cdot \mathbf{q}_R + (-\frac{1}{\theta}\mathbf{q}_R + \partial_{\mathbf{q}_R}\psi_R \mathbf{K}_R) \cdot \nabla_R \theta = \rho_R \theta \tilde{\gamma} \geq 0,$$

where $\tilde{\gamma} = \gamma$ at $\dot{\theta} = 0$. The arbitrariness of $\nabla_R \theta$ implies

$$-\frac{1}{\theta}\mathbf{q}_R + \partial_{\mathbf{q}_R}\psi_R \mathbf{K}_R = \mathbf{0}, \qquad f \partial_{\mathbf{q}_R} \psi_R \cdot \mathbf{q}_R = \rho_R \theta \tilde{\gamma} \geq 0.$$

Consequently,

$$\psi_R = \frac{1}{2\theta}\mathbf{q}_R \cdot \mathbf{K}_R^{-1} \mathbf{q}_R + \breve{\psi}_R(\theta, \mathbf{E}, \nabla_R \theta). \tag{35}$$

Further, $\gamma$ is given by

$$\rho_R \theta \gamma = -\frac{f}{\theta}\mathbf{q}_R \cdot \mathbf{K}_R^{-1} \mathbf{q}_R - \alpha \dot{\theta}^2.$$

*6.1. Structure of $\varepsilon_R$ and $\eta_R$*

To understand the possible consequences of the dependence on $\nabla_R \theta$ and $\dot{\theta}$ we determine the explicit form of the internal energy $\varepsilon$ and the entropy $\eta_R$. Indeed, for simplicity and definiteness we specify the free energy $\psi_R$ in (35) in the form

$$\psi_R = \frac{1}{2\theta}\mathbf{q}_R \cdot \mathbf{K}_R^{-1} \mathbf{q}_R + \Psi(\theta, \mathbf{E}) + \tfrac{1}{2}\beta|\nabla_R \theta|^2,$$

where $\beta$ is constant. Hence, we have

$$\eta_R = -\partial_\theta \psi_R + \frac{\beta}{\theta}|\nabla_R \theta|^2 - \beta \Delta_R \theta - \alpha \dot{\theta}$$

where $\Delta_R \theta = \nabla_R \cdot \nabla_R \theta$ and

$$\partial_\theta \psi_R = -\frac{1}{2\theta^2}\mathbf{q}_R \cdot \mathbf{K}_R^{-1} \mathbf{q}_R + \partial_\theta \Psi(\theta, \mathbf{E}).$$

Hence, we find

$$\varepsilon = \psi_R + \theta(-\partial_\theta \psi_R - \frac{\beta}{\theta}|\nabla_R \theta|^2 + \beta \Delta_R \theta - \alpha \dot{\theta}).$$

We then compute $\dot{\varepsilon}_R$ to obtain

$$\dot{\varepsilon}_R = \partial_{\mathbf{q}_R} \psi_R \cdot \dot{\mathbf{q}}_R + \partial_{\mathbf{E}} \psi_R \cdot \dot{\mathbf{E}} + \beta \nabla_R \theta \cdot \nabla_R \dot{\theta} - \frac{\beta}{\theta} \dot{\theta} |\nabla_R \theta|^2 + \beta \dot{\theta} \Delta_R \theta - \alpha \dot{\theta}^2 - \theta \overline{\partial_\theta \psi_R}$$
$$+ \beta \frac{\dot{\theta}}{\theta} |\nabla_R \theta|^2 - 2\beta \nabla_R \theta \cdot \nabla_R \dot{\theta} + \beta \theta \Delta_R \dot{\theta} - \alpha \theta \ddot{\theta}. \qquad (36)$$

Relative to the variables $\theta$, $\mathbf{E}$ and $\mathbf{q}_R$, by (36) we see that $\dot{\varepsilon}$ encloses first- and second-order derivatieves and furthermore a third-order term, $\beta \theta \Delta_R \dot{\theta}$.

### 6.2. Discontinuity Waves

To investigate the existence of thermal-mechanical waves we consider the balance equations and the rate equation for $\mathbf{q}_R$ in the form

$$\begin{aligned}
\rho_R \dot{\mathbf{v}} &= \nabla_R \cdot \mathbf{T}_R + \rho_R \mathbf{b}, \\
\dot{\varepsilon}_R &= \mathbf{T}_R \cdot \dot{\mathbf{F}} - \nabla_R \cdot \mathbf{q}_R + \rho_R r, \\
\dot{\mathbf{q}}_R &= -f \mathbf{q}_R - \mathbf{K}_R \nabla_R \theta,
\end{aligned} \qquad (37)$$

where $\mathbf{T}_R = \mathbf{F}\mathbf{T}_{RR}$ is the first Piola stress. We look for discontinuity waves ([1,19], Chapter 2) where

$$[\![\mathbf{T}_R]\!] = \mathbf{0}, \quad [\![\mathbf{q}_R]\!] = \mathbf{0}, \quad [\![\mathbf{v}]\!] = \mathbf{0}, \quad [\![\mathbf{F}]\!] = \mathbf{0}, \quad [\![\theta]\!] = 0.$$

Hence, by (37) we can write the system

$$\begin{aligned}
\rho_R [\![\dot{\mathbf{v}}]\!] &= [\![\nabla_R \cdot \mathbf{T}_R]\!], \\
[\![\dot{\varepsilon}_R]\!] &= \mathbf{T}_R \cdot [\![\dot{\mathbf{F}}]\!] - [\![\nabla_R \cdot \mathbf{q}_R]\!], \\
[\![\dot{\mathbf{q}}_R]\!] &= -\mathbf{K}_R [\![\nabla_R \theta]\!].
\end{aligned} \qquad (38)$$

Let $U$ be the (normal) speed and $\mathbf{n}_R$ the unit normal of the wave. Using the geometrical–kinematical conditions of compatibility we can write the system (38) in the form

$$\begin{aligned}
\rho_R U [\![\dot{\mathbf{v}}]\!] &= -[\![\dot{\mathbf{T}}_R]\!] \mathbf{n}_R, \\
U [\![\dot{\varepsilon}_R]\!] &= -(\mathbf{T}_R \mathbf{n}_R) \cdot [\![\dot{\mathbf{v}}]\!] + [\![\dot{\mathbf{q}}_R]\!] \cdot \mathbf{n}_R, \\
U [\![\dot{\mathbf{q}}_R]\!] &= \mathbf{K}_R \mathbf{n}_R [\![\dot{\theta}]\!].
\end{aligned} \qquad (39)$$

We notice that, except for $[\![\dot{\varepsilon}]\!]$, all the jumps are linear in the sought discontinuities. Instead $\dot{\varepsilon}$ involves the term $-\beta \Delta_R \dot{\theta}$, a single third-order derivative of $\theta$. Consequently this scheme is not consistent with propagation at a finite speed. To overcome this drawback we assume $\beta = 0$, which means that $\partial_{\nabla_R \theta} \psi_R = \mathbf{0}$. Hence, accounting for a dependence of the free energy on $\nabla_R \theta$ is not consistent with the propagation at a finite wave speed.

The dependence of the entropy on $\dot{\theta}$ results in a single second-order term in $\dot{\varepsilon}$, i.e., $-\alpha \theta \ddot{\theta}$. Hence, the compatibility with a finite wave speed also requires that $\alpha = 0$. We are then confined to the thermoelastic solid with a MC-like equation for the heat flux.

### 6.3. Waves in Thermoelastic Solids

Letting $\alpha = 0, \beta = 0$ we have

$$\psi_R = \frac{1}{2\theta} \mathbf{q}_R \cdot \mathbf{K}_R^{-1} \mathbf{q}_R + \Psi(\theta, \mathbf{E}), \qquad \eta_R = \frac{1}{2\theta^2} \mathbf{q}_R \cdot \mathbf{K}_R^{-1} \mathbf{q}_R - \partial_\theta \Psi(\theta, \mathbf{E}).$$

Hence, we obtain

$$\dot{\varepsilon}_R = \mathbf{T}_R \cdot \dot{\mathbf{F}} - \left( \frac{1}{\theta^2} \mathbf{q}_R \cdot \mathbf{K}_R^{-1} \mathbf{q}_R + \theta \partial_\theta^2 \Psi \right) \dot{\theta} + \frac{2}{\theta} (\mathbf{K}_R^{-1} \mathbf{q}_R) \cdot \dot{\mathbf{q}}_R - \theta (\partial_\theta \mathbf{T}_{RR}) \cdot \dot{\mathbf{E}}.$$

For simplicity we assume $\partial_\theta \mathbf{T}_{RR} = \mathbf{0}$, which is true if, e.g., $\Psi(\theta, \mathbf{E}) = \Psi_1(\theta) + \Psi_2(\mathbf{E})$. Upon substitution of $\dot{\varepsilon}_R$, the second equation in the system $(39)_2$ is given the explicit form

$$U\{\lambda[\![\dot\theta]\!] + \mathbf{p} \cdot [\![\dot{\mathbf{q}}_R]\!]\} = [\![\dot{\mathbf{q}}_R]\!] \cdot \mathbf{n}_R,$$

where

$$\lambda = -\theta \partial_\theta^2 \Psi - \frac{1}{\theta^2} \mathbf{q}_R \cdot \mathbf{K}_R^{-1} \mathbf{q}_R, \qquad \mathbf{p} = \frac{2}{\theta}(\mathbf{K}_R^{-1} \mathbf{q}_R).$$

We notice that $-\theta \partial_\theta^2 \Psi$ is the classical (positive) specific heat, and hence $\lambda$ is an effective specific heat. Substitution of $\dot{\mathbf{q}}_R$ from $(39)_3$ yields

$$\lambda U^2 + (\mathbf{p} \cdot \mathbf{K}_R \mathbf{n}_R)U - \mathbf{n}_R \cdot (\mathbf{K}_R \mathbf{n}_R) = 0.$$

Hence, we find two possible speeds,

$$U_\pm = -\frac{\mathbf{p} \cdot \mathbf{K}_R \mathbf{n}_R}{2\lambda} \pm \Big[\frac{(\mathbf{p} \cdot \mathbf{K}_R \mathbf{n}_R)^2}{4\lambda^2} + \frac{\mathbf{n}_R \cdot \mathbf{K}_R \mathbf{n}_R}{\lambda}\Big]^{1/2},$$

for the propagation of the discontinuity $[\![\dot\theta]\!]$.

Meanwhile, Equation $(39)_1$ results in the classical equation for the mechanical discontinuity $[\![\dot{\mathbf{v}}]\!]$ of acceleration waves [19].

Some comments are in order. The independent behaviour of $[\![\dot{\mathbf{v}}]\!]$ and $[\![\dot\theta]\!]$ is a consequence of the model; here it follows from the neglect of $\partial_\theta \mathbf{T}_{RR}$ in $(39)_2$. For thermodynamic consistency, the modelling of heat conduction through an MC-like equation affects the rate $\dot{\varepsilon}_R$ in that the free energy is required to depend on the heat flux. A family of models, with different physical properties, is obtained by letting $\mathbf{K}_R$ depend on temperature, while here $\mathbf{K}_R$ is constant for simplicity.

## 7. On a Rate-Dependent Theory of Heat Conduction in Rigid Solids

In an attempt to account for the propagation of thermal waves without any recourse for the rate-type MC equation, Bogy and Naghdi [20] considered the triplet $(\theta, \nabla\theta, \dot\theta)$ as the set of variables for the model of a rigid heat conductor. We now briefly review this possibility based on the energy equation

$$\rho\dot{\varepsilon} = -\nabla \cdot \mathbf{q} + \rho r \tag{40}$$

and the CD inequality

$$-\rho(\dot\psi + \eta\dot\theta) - \frac{1}{\theta^2} \mathbf{q} \cdot \nabla\theta = \rho\gamma \geq 0.$$

If $\psi, \eta, \mathbf{q}$ and $\gamma$ are functions of $(\theta, \nabla\theta, \dot\theta)$ then the CD inequality can be written in the form

$$-\rho(\partial_\theta \psi + \eta)\dot\theta - \rho\partial_{\nabla\theta}\psi \cdot \nabla\dot\theta - \rho\partial_{\dot\theta}\psi \ddot\theta - \frac{1}{\theta^2} \mathbf{q} \cdot \nabla\theta = \rho\gamma \geq 0.$$

The arbitrariness of $\nabla\dot\theta$ and $\ddot\theta$ implies that

$$\partial_{\nabla\theta}\psi = \mathbf{0}, \qquad \partial_{\dot\theta}\psi = 0$$

and hence $\psi = \psi(\theta)$ and the CD inequality simplifies to

$$-\rho(\partial_\theta \psi + \eta)\dot\theta - \frac{1}{\theta^2} \mathbf{q} \cdot \nabla\theta = \rho\gamma \geq 0.$$

More detailed consequences follow on the basis of some assumptions. Let

$$\eta(\theta, \nabla\theta, \dot\theta) = \eta_0(\theta) + \eta_c(\theta, \nabla\theta, \dot\theta), \qquad \eta_c(\theta, \mathbf{0}, \dot\theta) \to 0 \text{ as } \dot\theta \to 0,$$

$$\mathbf{q}(\theta, \nabla\theta, \dot\theta) = \mathbf{q}_0(\theta, \mathbf{0}, \dot\theta) + \mathbf{q}_c(\theta, \nabla\theta, \dot\theta), \qquad \mathbf{q}_c(\theta, \nabla\theta, 0) \to \mathbf{0} \text{ as } \nabla\theta \to \mathbf{0}.$$

We can then write the inequality in the form

$$-\rho(\partial_\theta\psi + \eta_0)\dot\theta - \rho\eta_c\dot\theta - \frac{1}{\theta^2}\mathbf{q}_0\cdot\nabla\theta - \frac{1}{\theta^2}\mathbf{q}_c\cdot\nabla\theta = \rho\gamma \geq 0.$$

Hence, as $\nabla\theta = \mathbf{0}$ we obtain

$$\eta_0 = -\partial_\theta\psi, \qquad \eta_c(\theta, \mathbf{0}, \dot\theta)\dot\theta \leq 0,$$

while as $\dot\theta = 0$ we find

$$\mathbf{q}_0 = \mathbf{0}, \qquad \mathbf{q}_c(\theta, \nabla\theta, 0)\cdot\nabla\theta \leq 0.$$

*Plane Thermal Waves*

We can investigate plane waves by restricting attention to a one-dimensional setting $(\mathbf{q}_c \to q_c, \nabla\theta \to \partial_x\theta)$. Using Lagrange's formula we can write

$$\eta_c(\theta, \mathbf{0}, \dot\theta) = \partial_{\dot\theta}\eta_c(\theta, 0, \xi)\dot\theta, \quad \xi \in (0, \dot\theta),$$

$$q_c(\theta, \partial_x\theta, 0) = \partial_{\partial_x\theta}q_c(\theta, \zeta, 0)\partial_x\theta, \quad \zeta \in (0, \partial_x\theta).$$

Necessary conditions are $\partial_{\dot\theta}\eta_c(\theta, 0, \dot\theta) \leq 0$ and $\partial_{\partial_x\theta}q_c(\theta, \partial_x\theta, 0) \leq 0$. We assume the sufficient conditions

$$\partial_{\dot\theta}\eta_c(\theta, \partial_x\theta, \dot\theta) < 0, \quad \partial_{\partial_x\theta}q_c(\theta, \partial_x\theta, \dot\theta) < 0$$

are valid in a neighbourhood of the origin.

Since

$$\dot\varepsilon = (\eta_c + \theta\partial_\theta\eta_c - \partial_\theta^2\psi)\dot\theta + \theta\partial_{\partial_x\theta}\eta_c\,\partial_x\dot\theta + \theta\partial_{\dot\theta}\eta_c\,\ddot\theta,$$

$$\partial_x q = \partial_\theta q\partial_x\theta + \partial_{\partial_x\theta}q\,\partial_x^2\theta + \partial_{\dot\theta}q\partial_x\dot\theta,$$

then Equation (40) can be written in the form

$$a\partial_x^2\theta + p\partial_x\dot\theta + b\ddot\theta + f\partial_x\theta + g\dot\theta = \rho r,$$

where

$$a = \partial_{\partial_x\theta}q, \quad b = \theta\partial_{\dot\theta}\eta_c, \quad p = \partial_{\dot\theta}q + \rho\theta\partial_{\partial_x\theta}\eta_c.$$

The positive product $ab$ then makes the differential equation of the elliptic character not in favour of a model for wave propagation.

## 8. Conclusions

This paper investigates the techniques associated with the exploitation of the second law of thermodynamics as a restriction on physically admissible processes. As is standard in the literature, the exploitation consists of the use of arbitrariness occurring in the CD inequality. The present approach emphasizes two uncommon features within the thermodynamic analysis: the representation formula and the entropy production.

There are cases where more terms of the CD inequality are not independent. The representation formulae, for vectors or tensors, allow us to derive a direct dependence between appropriate unknowns. As an example, Equation (17) has the form

$$\boldsymbol{\mathcal{A}}\cdot\dot{\mathbf{E}} + \boldsymbol{\mathcal{B}}\cdot\dot{\mathbf{T}}_{RR} = \sigma$$

and the representation formula allows the unknown $\dot{\mathbf{T}}_{RR}$ to be determined. The solution is widely non-unique and this results in a variety of models characterized by the free energy $\psi$ and the right-hand side $\sigma = \rho_R\theta\gamma$. Among the examples developed in this paper, we obtained the constitutive equation for hypo-elastic solids and for MC-like equations of heat conduction. Further models can be established by using the techniques developed in this paper, as was performed in [10].

Concerning the entropy production $\gamma$, we let it be given by a constitutive function per se and not merely by the expression inherited by other constitutive functions. This property results in more general expressions of the representation formulae and, as shown in [7–9]), is crucial for the description of hysteretic phenomena.

These features are highlighted in this paper through models of viscoelastic solids and heat conductors. In particular, the models of heat conduction were also investigated in detail in connection with wave propagation properties.

**Author Contributions:** Investigation: A.M. and C.G. All authors have contributed substantially and equally to the work reported. All authors have read and agreed to the published version of the manuscript.

**Funding:** This work received no external funding.

**Institutional Review Board Statement:** Not applicable.

**Informed Consent Statement:** Not applicable.

**Data Availability Statement:** The study did not report any data.

**Acknowledgments:** The research leading to this work has been developed under the auspices of INDAM-GNFM.

**Conflicts of Interest:** The author declares no conflict of interest.

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
