# Peer review of "Techniques for the Thermodynamic Consistency of Constitutive Equations"

_2673-7264, doi:10.3390/thermo3020016_

Round 1

Reviewer 1 Report

I agree with the authors that the approach in thermodynamics by Green and Naghdi -although important to the reviewer's opinion - has not received much attention.

I suggest that the authors define the variational derivative introduced in Eq. (15).

I also suggest that the authors define the tensor product appearing before and in Eq. (27).

There are minor problems with respect to English. The authors should check the use of the appropriate preposition each time.

Author Response

Many thanks  to the reviewer. See the attachment. 

Reviewer 2 Report

At mesoscopic and macroscopic level, processes not allowed by the second law of thermodynamics  are not seen to occur in nature, and therefore they are rightly considered un-physical in spite they may comply with other laws, such as the conservation of energy and momentum. The second law can be brought to the form of the so-called Clausius-Duhem inequality which, being equivalent to the assertion that the entropy production cannot be negative, sets a restriction on physical, real, processes.  Methods to  take advantage of this restriction are the subject of the paper under consideration. The authors, deal in detail with visco-elasticity (section 4) and heat flow (sections 5-7 ) processes to exemplify  the said methods.

I find the paper a bit too formal but interesting anyway. This is why  I  favour publication.

Author Response

Many thanks for the comments. See the attachment.
